# Does DPP-IV Inhibition Offer New Avenues for Therapeutic Intervention in Malignant Disease?

**DOI:** 10.3390/cancers14092072

**Published:** 2022-04-21

**Authors:** Petr Busek, Jonathan S. Duke-Cohan, Aleksi Sedo

**Affiliations:** 1Laboratory of Cancer Cell Biology, Institute of Biochemistry and Experimental Oncology, First Faculty of Medicine, Charles University, 128 53 Prague, Czech Republic; 2Department of Medical Oncology, Dana-Farber Cancer Institute and Department of Medicine, Harvard Medical School, Boston, MA 02115, USA; jonathan_duke-cohan@dfci.harvard.edu

**Keywords:** gliptin, cancer, tumor microenvironment, immune response, chemokine, stromal cell-derived factor, drug repurposing, stem cells

## Abstract

**Simple Summary:**

There is growing interest in identifying the effects of antidiabetic agents on cancer risk, progression, and anti-cancer treatment due to the long-term use of these medications and the inherently increased risk of malignancies in diabetic patients. Tumor development and progression are affected by multiple mediators in the tumor microenvironment, several of which may be proteolytically modified by the multifunctional protease dipeptidyl peptidase-IV (DPP-IV, CD26). Currently, low-molecular-weight DPP-IV inhibitors (gliptins) are used in patients with type 2 diabetes based on the observation that DPP-IV inhibition enhances insulin secretion by increasing the bioavailability of incretins. However, the DPP-IV-mediated cleavage of other biopeptides and chemokines is also prevented by gliptins. The potential utility of gliptins in other areas of medicine, including cancer, is therefore being evaluated. Here, we critically review the existing evidence on the role of DPP-IV inhibitors in cancer pathogenesis, their potential to be used in anti-cancer treatment, and the possible perils associated with this approach.

**Abstract:**

Dipeptidyl peptidase IV (DPP-IV, CD26) is frequently dysregulated in cancer and plays an important role in regulating multiple bioactive peptides with the potential to influence cancer progression and the recruitment of immune cells. Therefore, it represents a potential contributing factor to cancer pathogenesis and an attractive therapeutic target. Specific DPP-IV inhibitors (gliptins) are currently used in patients with type 2 diabetes mellitus to promote insulin secretion by prolonging the activity of the incretins glucagon-like peptide 1 (GLP-1) and glucose-dependent insulinotropic polypeptide (GIP). Nevertheless, the modulation of the bioavailability and function of other DPP-IV substrates, including chemokines, raises the possibility that the use of these orally administered drugs with favorable side-effect profiles might be extended beyond the treatment of hyperglycemia. In this review, we critically examine the possible utilization of DPP-IV inhibition in cancer prevention and various aspects of cancer treatment and discuss the potential perils associated with the inhibition of DPP-IV in cancer. The current literature is summarized regarding the possible chemopreventive and cytotoxic effects of gliptins and their potential utility in modulating the anti-tumor immune response, enhancing hematopoietic stem cell transplantation, preventing acute graft-versus-host disease, and alleviating the side-effects of conventional anti-tumor treatments.

## 1. Introduction

There is a growing appreciation that the tumor microenvironment represents a critical parameter for the development and progression of malignancies. During the course of tumorigenesis, a tumor niche defined by and specific for the tissue in which the tumorigenic process was initiated is established and contributes to the tumor’s development and growth [1]. The recruitment of immune and other cell types into this niche and their interactions within it are dependent upon a complex display of chemokines and other signaling mediators. Concomitantly, a number of these mediators are involved in the maintenance of niches that are critical for tissue homeostasis and recovery after the stress induced by conventional anticancer treatments. Consequently, our knowledge of the control of the expression and activity of these mediators may open up novel or complementary avenues for intervention in malignant diseases.

The enzymatic activity of dipeptidyl peptidase IV (DPP-IV) has been established as a critical regulator of chemokines, neuropeptides, and complex metabolic regulators such as incretins [2,3,4,5,6]. In this article, we examine the possible impact of DPP-IV enzymatic activity inhibition in tumor cells and supporting niche stromal cells on tumor development, progression, and therapy. We review the currently available evidence suggesting the potential chemopreventive and therapeutic effects of DPP-IV inhibition, including its possible exploitation in boosting the anti-tumor immune response. We further discuss the possible limitations and risks stemming from the increased bioavailability of DPP-IV substrates in tumor niches and/or so far rather poorly understood other effects of DPP-IV inhibition, such as the activation of the nuclear factor E2–related factor 2 (Nrf2)-mediated antioxidant response [7,8].

## 2. DPP-IV (CD26)—A Multifunctional Molecule with Enzyme-Activity-Dependent and -Independent Functions

Dipeptidyl peptidase IV (DPP-IV, DPP4, CD26, adenosine deaminase complexing protein 2, EC 3.4.14.5) is a homodimeric type II transmembrane glycoprotein consisting of a short cytoplasmic region (amino acids 1–6), a 22-amino-acid-long transmembrane domain and a large extracellular domain (amino acids 29–766) with the catalytic triad formed in the human protein by Ser^630^, Asp^708^, and His^740^ [5]. In addition to the membrane-bound form that is expressed in a wide spectrum of cell types, including epithelial, endothelial, and immune cells, a soluble isoform lacking the 30–38 amino acids at the N terminus is present in bodily fluids such as blood plasma, saliva, cerebrospinal fluid, and seminal plasma [5,9].

Together with its homologues fibroblast activation protein (FAP), DPP8, and DPP9, DPP-IV belongs to the S9b DPP family of serine proteases [10]. Its proteolytic activity cleaves an X-Pro or X-Ala dipeptide from the N terminus of various peptides and proteins comprising a large group of bioactive molecules with pleiotropic biological functions (Table 1, see [2,3,4,5,6] for review). This cleavage results in changed receptor preference, activation, conversion to a receptor antagonist, or the inactivation of the bioactive molecule [3,11]. For example, DPP-IV was identified as an enzyme contributing to the regulation of insulin release by rapidly inactivating the incretins GLP-1 and GIP. These peptides are produced in the gastrointestinal tract in response to food intake and stimulate glucose-stimulated insulin secretion [12]. Soon after the discovery of this glucoregulatory mechanism, several potent low molecular weight competitive DPP-IV inhibitors (gliptins, *gli-* “antihyperglycemic” + *(pe)pt(idase)* + *in(hibitor)*, Table 2) were introduced into clinical practice for the treatment of type 2 diabetes mellitus (see [2] for review). Regardless of their rather modest efficacy, gliptins represent a frequently used option for diabetic patients due to their ease of use, minimal risk of hypoglycaemia, neutral effect on weight gain and cardiovascular events, and favorable side-effect profile [13,14]. Several studies further suggest that, most likely by increasing the bioavailability of GLP-1 and other DPP-IV substrates, gliptins not only improve hyperglycemia but may also have renoprotective, hepatoprotective, antioxidative, and anti-inflammatory effects [14,15,16,17,18,19,20], which seems favorable in the context of various pathological states closely associated with type 2 diabetes.

Other, less well understood, processes involving DPP-IV-mediated cleavage include the egress and homing of progenitor cells in the bone marrow regulated by CXCL12 (stromal-cell-derived factor 1, SDF-1) and other chemokines [26,27,28] and, possibly, the fine-tuning of satiety and food intake mediated by the cleavage of neuropeptides such as peptide YY (PYY) and NPY [29].

In addition to its proteolytic functions, DPP-IV engages in various non-hydrolytic protein–protein interactions. For example, the binding of viral proteins such as the HIV TAT [30], gp-120 [31] and MERS-CoV spike proteins [32] to DPP-IV has been suggested to play a role in the pathogenesis of the respective viral diseases. In addition, DPP-IV strongly binds human adenosine deaminase 1 (ADA), mannose 6-phosphate/insulin-like growth factor II receptor, caveolin-1, CARMA1, the chemokine receptor CXCR4, the extracellular matrix proteins vitronectin, collagen, and fibronectin, the plasminogen receptor, and the Na^+^/H^+^ exchanger isoform NHE3 (see [2,3] and references therein). 

Through non-hydrolytic protein–protein interactions and most likely also by its enzymatic activity, DPP-IV seems to play a role in the immune system. DPP-IV (more commonly referred to as CD26 in the field of immunology) was initially identified as marking activated and proliferating T cells [33]. Within T cells, the highest expression in humans is in mucosal-associated invariant T (MAIT) cells, responding to the bacterial riboflavin metabolites presented by MR1 [34]. In addition to MAIT, CD4^+^ and CD8^+^ effector memory cells (including the Th17 cells involved in autoimmune inflammatory reactions) and NKT cells tend to have high DPP-IV/CD26 levels, while naïve CD4^+^CD45RA^+^, CD8^+^CD45RA^+^, and CD4^+^CD25^+^ T_reg_ have low to moderate levels [35,36]. Mobilized hematopoietic stem cells (HSC) also express DPP-IV/CD26 [28]. Functional studies have demonstrated that CD26 binds caveolin-1 on antigen-presenting cells and, by upregulating the costimulatory molecule CD86, stimulates their ability to promote T-cell activation [37]. In addition, the caveolin-1 binding or crosslinking of DPP-IV (CD26) by antibodies induces T-cell activation and proliferation [38,39]. Developmentally, lymphocyte proliferation is tempered by apoptosis to generate an appropriate physiological response. Consequently, during initial amplification, apoptosis may need to be suppressed. Extracellular adenosine, which accumulates as cells proliferate, can bind to the A2a purinergic receptor on lymphocytes, initiating an antiproliferative response [40]. Proliferating human lymphocytes may be able to mitigate lymphotoxicity by sequestering adenosine deaminase (ADA) to the membrane, a function mediated by human CD26 that can bind ADA with high affinity [41]. Interactions between ADA and CD26 may further contribute to the adhesion of T cells to epithelial cells [42]. While the aforementioned effects seem to be largely independent of enzymatic activity [43,44], it is likely that cell-surface DPP-IV in immune cells may be instrumental in proteolytically modifying the activity of chemokines to ensure that once these cells have been recruited to an inflammatory site, their influx is limited by the inactivation of the local chemokine gradient [45]. Further, chemokine activity may be limited to the initiating inflammatory sites through high endothelial surface DPP-IV/CD26 representation in the vascular bed and significant activity of the secreted form of DPP-IV/CD26 within the circulation. 

A degree of controversy exists regarding the importance of DPP-IV/CD26 and its enzymatic activity in the immune response. Some studies demonstrate its significance [43,46,47], while others find little or no role [48,49]. The highly cell-type- and context-dependent function(s) of DPP-IV/CD26 may perhaps be redundant for the overall immune response, but they may become critical for the fine-tuning of its specific steps. Another important consideration is that DPP-IV/CD26 is differentially expressed between mice and humans. Unlike murine hematopoietic cells, in which CD26 is expressed well on thymocytes and mature T cells, dendritic cells, and B cells, in humans it is only expressed well on T cells in vivo [33,35,50]. In addition, only human, and not mouse DPP-IV/CD26 binds ADA [3], and the role of DPP-IV in the adhesion to fibronectin has been demonstrated for rodent [51] but, so far, not for human DPP-IV. These differences need to be taken into account when extrapolating from the results of animal studies.

In summary, DPP-IV seems to affect a host of biological processes in the body by non-proteolytic protein–protein interactions and by locally and systemically regulating the availability of bioactive signaling molecules. The use of gliptins in diabetic patients may therefore impact on several areas, including the development and progression of tumors. On the other hand, DPP-IV inhibition may prove beneficial in certain situations in patients with malignant diseases, and the availability of highly specific and approved DPP-IV inhibitors opens up the possibility of their repurposing in oncology.

## 3. DPP-IV in Cancer—A Matter of Controversy

Based on its differential expression in tumors and corresponding normal tissues, the role of DPP-IV in cancer pathogenesis was hypothesized very soon after its discovery. Numerous studies demonstrated the potential of DPP-IV to act either as a tumor promoter or as a tumor suppressor. In addition, the expression of DPP-IV in cancer tissue, as well as its levels in body fluids, were considered possible biomarkers in a range of human malignancies (reviewed in [52,53,54]). In addition to its roles in cell adhesion and immune cell signaling, which are typically executed by non-enzymatic mechanisms, DPP-IV enzymatic activity processes multiple regulatory peptides and, thus, modifies the message conveyed by them to both DPP-IV-expressing and DPP-IV-non-expressing cells present in the tumor microenvironment [11]. 

The tumor-suppressing effect of DPP-IV has been postulated in several cancers based on experiments modulating DPP-IV expression in cancer cell line models. The transgenic overexpression of DPP-IV abrogated cell growth, migration, resistance to apoptosis, and metastasis in non-small-cell lung [55], neuroblastoma [56], and melanoma cell lines [57]. The loss of DPP-IV expression was observed and a consequent decrease in CXCL12 degradation was postulated in Sezary syndrome. Moreover, the exogenous addition of soluble DPP-IV prevented the migration and skin homing of malignant cells, while DPP-IV inhibitors abolished these effects [58]. 

By contrast, a tumor-promoting effect of DPP-IV was suggested in pleural mesothelioma, where DPP-IV is expressed in cancer stem cells and might serve as a promising therapeutic target [59]. The association of high DPP-IV expression with BRAF mutation, cancer cell invasion, and a more aggressive course of disease was observed in papillary thyroid cancer [60]. In gastric cancer, a subpopulation of cancer stem cells defined by DPP-IV and CXCR4 expression exhibited higher migration, invasion, and clonogenic capacity [61]. DPP-IV is upregulated in premalignant adenomas and colorectal carcinomas compared to normal human large intestine tissue [62]. In addition, a subpopulation of DPP-IV-positive colorectal cancer stem cells was described in primary tumors, as well as in their liver metastases. These DPP-IV-expressing cells seemed to be critical for tumor initiation, progression, and chemoresistance [63]. Further, an enzyme-activity-dependent role of DPP-IV in the process of colon cancer cell metastasis was demonstrated in a mouse model, in which DPP-IV inhibition blocked the growth of lung metastases [64]. The increased expression of DPP-IV and its association with the presence of distant metastases was reported in esophageal adenocarcinoma [65]. Increased DPP-IV expression, possibly contributing to liver cancer progression by converting NPY to the Y5 receptor agonist NPY(3-36) [66], was further associated with tumor size, stage, and proliferation capacity in hepatocellular cancer. In addition, the administration of DPP-IV inhibitors in mouse hepatocellular carcinoma models promoted anti-tumor immunity [67,68,69].

To draw a clear-cut conclusion regarding the resulting pro- or anti-tumorigenic role of DPP-IV in a number of other cancers is rather challenging. The increased expression of DPP-IV was observed in human prostate cancer tissue and, to lesser extent, in adjacent benign hyperplastic glands [70,71]. A small study demonstrated a positive correlation between DPP-IV expression and PSA level and cancer stage, suggesting that DPP-IV may represent a negative prognostic marker in prostate cancer [71]. However, DPP-IV was reported to act as a tumor suppressor in prostate adenocarcinoma cells, where its transgenic overexpression led to apoptosis, cell cycle arrest, and the inhibition of in vitro cell migration and invasion [72]. In support of these findings, other studies concluded that DPP-IV is an epigenetically regulated tumor suppressor in castration-resistant prostate cancer. DPP-IV inhibition may result in increased CXCL12-induced migration and invasion of transformed prostate cancer cells and contribute to castration resistance [73,74]. Similarly, DPP-IV expression is upregulated in high-grade astrocytic tumors [75], while our in vitro and in vivo studies suggested growth inhibitory effects, possibly involving non-enzymatic mechanisms [76,77]. Variable DPP-IV expression has been reported in normal and breast cancer tissues [78], but literature data are scarce [53]. In breast cancer cell lines and patient-derived cells, DPP-IV expression varied among different patient samples and histological subtypes [79]. Transgenic DPP-IV overexpression led to increased EGF-induced colony formation in normal breast epithelial cells and facilitated EGF signaling via MEK/ERK and JNK/c-Jun, leading to the expression of peptidylprolyl cis/trans isomerase-1 (PIN1) and cyclin D1 in cancer cells. This was correlatively confirmed in breast cancer biopsy material, where the presence of DPP-IV correlated with PIN1 expression. DPP-IV silencing, as well as activity inhibition, were shown to suppress EGF-induced epithelial transformation and mammary tumorigenesis; however, the specific DPP-IV substrate(s) involved in these processes were not identified [78]. By contrast, others observed an association between low DPP-IV, metastasis, and, possibly, resistance to therapy. Probably due to decreased CXCL12 cleavage, DPP-IV ablation promoted epithelial–mesenchymal transition, breast cancer metastasis, and the upregulation of ABC transporters via the CXCL12/CXCR4/mTOR axis [80,81]. In endometrial carcinoma, Yang et al. described a positive association between DPP-IV expression and cancer cell proliferation, invasion, and tumorigenicity. DPP-IV upregulation in endometrial cancer cells in vitro activated HIF-1alpha–VEGF signaling and stimulated the expression of proproliferative IGF-1. By contrast, DPP-IV knockdown and enzymatic activity inhibition decreased cancer cell growth [82]. The opposite conclusions were published by Khin et al., who observed a negative correlation between DPP-IV and tumor grade and speculated as to the possible antiproliferative effect of the enzyme, possibly due to the degradation of local proproliferative peptides such as the chemokine CCL5 (RANTES) [83]. The role of DPP-IV expression also remains unclear in ovarian cancer. On the one hand, ovarian cancer cells with low DPP-IV expression were more invasive, and DPP-IV overexpression led to increased adhesion in vitro and reduced peritoneal dissemination and growth in an animal model [84,85]. In addition, DPP-IV overexpression improved the sensitivity of the cancer cells to chemotherapy, an effect that was probably not dependent on the enzymatic activity [86]. On the other hand, an immunohistochemical study by other authors provided evidence of higher DPP-IV expression in ovarian cancer compared to benign ovarian tumors and an association between DPP-IV positivity and the presence of lymph-node metastases [87].

Together, the results of studies analyzing the role of DPP-IV in cancer are often ambiguous, suggesting a pro- or anti-oncogenic role depending on the cancer type and, possibly, other factors. A probable explanation for these seemingly contradictory observations may be derived from the broad spectrum and pleiotropic effects of the biologically active DPP-IV substrates [11]. Therefore, the microenvironment-specific context comprising DPP-IV in various cell types and the locally specific mix of its substrates, which eventually—through an autocrine and/or paracrine mechanism—creates a complex mosaic of net pro- or anti-oncogenic effects of the enzyme in a particular malignancy. The potential of DPP-IV inhibitors to disturb the levels of local regulatory peptides and possibly affect oncogenic processes in malignant and premalignant niches is thus of great interest. 

## 4. DPP-IV Inhibition and Cancer Initiation and Progression

Currently, more than ten potent and relatively selective DPP-IV inhibitors (gliptins) are used in clinical practice worldwide. The systemic inhibition of DPP-IV enzymatic activity with the ensuing increased bioavailability and signaling of various DPP-IV substrates, including those that may affect cell proliferation and migration, has raised concerns in two areas.

The first is its possible impact on diverse processes in premalignant niches, resulting in their more rapid evolution into cancer. The increased risk of pancreatic cancer ranks among the most frequently discussed and controversial topics related to the use of DPP-IV inhibitors. DPP-IV expression is upregulated in pancreatic adenocarcinoma, which may be linked to the impaired glucose homeostasis frequently associated with this type of cancer [88]. In a transgenic rat model of type 2 diabetes mellitus, DPP-IV inhibition by sitagliptin increased ductal cell proliferation and induced ductal cell metaplasia as a precursor of pancreatic ductal adenocarcinoma, possibly due to the increased availability of GLP-1 [89]. In line with these findings, an autopsy study in patients treated with sitagliptin reported a marked expansion of the exocrine and endocrine compartments of the pancreas [90]. By contrast, an independent study revealed no pronounced histological abnormalities in the pancreatic tissues of diabetic patients treated with incretins compared to diabetic patients not receiving incretins [91]. In addition, three different studies on rodents failed to reveal changes in ductal proliferation or evidence of pancreatitis, ductal metaplasia, or neoplasia with gliptin treatment lasting for up to two years [92,93,94]. Conflicting data on pancreatic cancer were also reported in clinical studies and meta-analyses, some of which showed no increase or even a numerical decrease in risk [95,96,97,98,99], while others reported increased risk [100,101]. A large population-based cohort study has also demonstrated an increased risk of cholangiocarcinoma in patients using gliptins [102], but mechanistic studies in this serious but relatively rare cancer type are lacking.

The second area of concern is that DPP-IV inhibition may support the growth/invasive spread of cancer cells in patients with a preexisting malignancy. A case study reported that a patient with a recurrent metastatic carcinoid tumor that was stable for several years experienced an almost twofold increase in plasma serotonin levels after the initiation of saxagliptin, suggesting tumor progression. Upon the discontinuation of the medication, the values quickly returned to their previous levels, implying causality [103]. A study in a mouse model of breast cancer suggested that the inhibition of DPP-IV may promote the epithelial–mesenchymal transition, proliferation, and metastasis of cancer cells through the CXC12–CXCR4 axis [80]. Although the inhibitor utilized in this study had a rather low potency [104] and is not used in clinical practice, this report raises a concern that other DPP-IV inhibitors may have a similar effect. In another study, using a mouse model of prostate cancer, sitagliptin administration led to a more rapid restoration of tumor growth after castration, suggesting that DPP-IV inhibition may decrease the efficacy of androgen deprivation therapy [74]. Saxagliptin and sitagliptin further promoted the migration and invasion of various cancer cell lines in vitro without affecting their proliferation or sensitivity to chemotherapeutics and promoted metastatic dissemination in vivo. This was linked to the reduced oxidative stress caused by the decreased ubiquitination and resulting activation of the nuclear factor E2–related factor 2 (Nrf2) antioxidative pathway in the cancer cells [7]. A similar Nrf2-mediated promigratory and proinvasive effect, and a more rapid growth of lung metastases, was observed for thyroid cancer cells [8]. DPP-IV inhibition is highly likely to be responsible for these effects, since several structurally diverse gliptins activated Nrf2. Nevertheless, the underlying molecular mechanism is currently unclear and the responsible DPP-IV substrate(s) remain(s) to be identified. The increased metastatic spread of a preexisting tumor, as observed in preclinical models, has so far not been reported in clinical studies analyzing the development of metastatic disease in type 2 diabetes patients treated with DPP-IV inhibitors. An observational German study with a 3–4 year follow-up did not reveal an increased risk of metastases in DPP-IV-inhibitor-treated patients with breast, prostate or digestive-organ cancers [105]. Similarly, a study with a 5.6-year follow-up in Korean diabetic patients with preexisting primary cancer did not reveal an association between the use of DPP-IV inhibitors and new metastatic disease. A possible exception was a subgroup of patients with thyroid cancer, but the overall frequency of metastatic disease in this small subgroup was low [106]. These studies in two different populations provide reassurance, but caution is advisable in interpreting the results given the inherent limitations of such studies and their relatively short follow-up. 

In contrast to the worrying reports mentioned above, several epidemiological studies suggest that gliptin use is associated with a reduced risk of breast [107], prostate [108], and HCV infection-associated hepatocellular cancer [109]. In addition, a small study reported improved outcomes in tyrosine-kinase-inhibitor-treated patients with renal cell cancer using gliptins [110]. Although the mechanisms remain largely unknown, the reduction in cancer initiation and/or progression by gliptins is supported by several preclinical studies (Table 3). DPP-IV is upregulated in hepatocellular carcinoma [66,111] and high serum DPP-IV activity is associated with worse patient survival [112]. In animal models, gliptins ameliorated chemically induced liver fibrosis [15,113,114,115,116], a known risk factor for hepatocellular carcinoma. In addition, vildagliptin prevented diethylnitrosamine-induced hepatocarcinogenesis in rats fed a high-fat diet. Increased vascularization and macrophage, but not the T- or NK-cell infiltration of hepatic nodules, together with lung metastases, were observed in animals fed a high-fat diet, and these effects were suppressed by vildagliptin. Interestingly, vildagliptin normalized the high-fat diet-induced elevation of plasma CCL2, a potential contributor to the proangiogenic effects. Similar results were obtained in DPP-IV knockout animals, suggesting a critical role of DPP-IV enzymatic activity in these processes [112]. Hepatoprotective effects were also observed for saxagliptin in a rat model of hepatic injury induced by thioacetamide [117]. Saxagliptin reduced the serum levels of the markers of hepatocellular injury and alpha-fetoprotein, diminished the histopathological changes induced by thioacetamide, and deferred the occurrence of hepatocellular carcinoma, possibly by suppressing Wnt/Hedgehog/Notch1 signaling. Of note, the protective effects were observed even when DPP-IV inhibition was initiated several weeks after exposure to hepatotoxic treatment [117]. However, whether and which of these effects are mediated by DPP-IV substrates affecting hepatocytes, immune cells [68], or both remains unknown. 

Nonalcoholic steatohepatitis (NASH) is an increasingly prevalent disease, which leads to liver cirrhosis and predisposes patients to hepatocellular carcinoma. DPP-IV may participate in the pathogenesis of NASH by regulating the levels of bioactive GLP-1, but also through its auto and paracrine effects on hepatic insulin signaling [118]. In a genetically obese melanocortin 4 receptor (MC4R)-deficient mouse model of NASH, anagliptin ameliorated fibrosis and reduced the number of liver tumors that developed after the mice were fed a Western-type diet. Anagliptin had no effect on body weight, systemic glucose and lipid metabolism, hepatic steatosis, or adipose tissue inflammation, but its protective effects were proposed to be mediated by the GLP-1-mediated suppression of macrophage activation [119]. Concordantly, sitagliptin mitigated hepatic steatosis and reduced the number and size of hepatic tumors developing in a STAM mouse model of NASH. The authors proposed that these effects may be linked to the inhibition of the pentose phosphate pathway in the tumor tissue, possibly resulting from the suppression of the p62–Keap1–Nrf2 pathway [120]. Two studies utilizing a choline deficiency-induced steatohepatitis rat model [121,122] demonstrated that gliptins may reduce the development of liver fibrosis and preneoplastic lesions; this is likely to occur through their inhibition of activated stellate cells, through their role in reducing oxidative stress, and through their inhibition of angiogenesis. The combination with either a sodium-glucose cotransporter-2 (SGLT2) inhibitor [121] or an angiotensin II receptor type 1 (AT1) antagonist [122] was synergistic and led to more pronounced protective effects.

The chemopreventive effects of gliptins were also reported in colorectal cancer, about which DPP-IV inhibition has raised concerns because of its potential to enhance the intestinotrophic effects of GLP-2 [123,124]. Despite this, no association was observed between gliptin use and the incidence of colorectal carcinoma [125]. In fact, diabetic patients with colorectal carcinoma treated with gliptins seem to have improved outcomes compared to those treated with other hypoglycemic medications [126,127,128]. In addition, several models emulating various aspects of intestinal tumorigenesis show rather protective effects of gliptins. High-fat diet-fed rats exposed to the carcinogen 1,2-dimethylhydrazine developed lower numbers of precancerous lesions with sitagliptin administration [129]. Lower blood plasma levels of hydrogen peroxide in animals receiving sitagliptin were reported in this study [129], but whether and how this related to DPP-IV inhibition is currently unclear. A similar study with sitagliptin in leptin-deficient mice, in which intestinal carcinogenesis was initiated using 1,2-dimethylhydrazine together with chemically induced colitis, reported a lower number of aberrant crypt foci and lower intestinal IL6 expression [130]. Further, rather than an increase, a statistically nonsignificant decrease in tumor number was described for sitagliptin in a model of high-fat diet-induced carcinogenesis in mice carrying a heterozygous mutation in the adenomatous polyposis coli (*Apc*) tumorsuppressor gene (C57BL/6J-*Apc^Min^*/J) [131]. Somewhat counterintuitively, the long-term sitagliptin administration in this study lowered the high-fat diet-induced increase in GLP-2, CXCL5, and CXCL12 in the blood plasma [131]. 

Preclinical data on other types of cancer are scarce. A study using a model of chemically induced renal cell carcinoma showed that, probably by improving the defense against oxidative stress and ameliorating inflammation, sitagliptin decreased the occurrence of neoplastic foci in the kidney cortex and improved renal functions [132]. Sitagliptin further reduced tumor growth in transgenic mice that expressed oncogenically activated MAPK kinase 1 in non-dividing, differentiating epidermal cells, in which skin wounding induced tumor formation. Nevertheless, the onset, formation, and number of tumors were similar, and there was no effect of sitagliptin on angiogenesis or cancer cell proliferation [133].

Collectively, current evidence from clinical studies regarding the effect of gliptins on cancer development and progression is rather conflicting [107,108,109,125,126,127,128,134,135,136,137], possibly due to the long period during which most malignancies develop before they are diagnosed and the complexity of the pathogenetic mechanisms determining their growth, invasion, and metastatic spread. Preclinical studies suggest that gliptins may impede the development of certain tumors, such as hepatocellular carcinoma. On the other hand, there may be a risk that DPP-IV inhibition may encourage the progression of some preexisting malignancies by enhancing the spread of cancer cells. 

**Table 3 cancers-14-02072-t003:** Summary of preclinical studies demonstrating possible chemopreventive effects of gliptins.

Tumor Type	Model	Gliptin	Reference
Hepatocellular carcinoma	Diethylnitrosamine + high-fat diet-induced carcinogenesis in rats	Vildagliptin	[112]
	Thiacetamide-induced carcinogenesis in rats	Saxagliptin	[117]
Hepatocellular carcinoma associated with nonalcoholic steatohepatitis	Melanocortin 4 receptor (MC4R)-deficient mice fed Western-type diet	Anagliptin	[119]
	STAM mouse model	Sitagliptin	[120]
	Choline deficiency-induced steatohepatitis in rats	Teneligliptin	[121]
	Choline deficiency-induced steatohepatitis in rats	Sitagliptin	[122]
Colorectal cancer	1,2-dimethylhydrazine and high-fat diet in rats	Sitagliptin	[129]
	Leptin-deficient mice administered 1,2-dimethylhydrazine and dextran sulfate sodium-induced colitis	Sitagliptin	[130]
	Mice with heterozygous Apc mutation fed high-fat diet	Sitagliptin	[131]
Renal cell carcinoma	Diethylnitrosamine-induced carcinogenesis in rats	Sitagliptin	[132]

## 5. DPP-IV Inhibition in Anticancer Treatment

### 5.1. Direct Cytotoxic Effects of DPP-IV Inhibition on Cancer Cells

Based on the overexpression of DPP-IV in the tumor microenvironment and its proposed tumor-promoting role in certain cancer cells, several studies investigated whether gliptins have a direct cytotoxic effect on cancer cells (Table 4). Vildagliptin has been shown to decrease the growth of colorectal cancer cells in vitro [64]. In addition, it inhibited the growth of lung metastases in a mouse model, possibly by promoting cancer cell apoptosis and decreasing autophagy [64]. In a series of three closely related in vitro studies, the cytotoxic effects of a highly selective DPP-IV inhibitor, gemigliptin, were evaluated in thyroid cancer cells by Kim et al. The authors reported that gemigliptin exhibited a dose- and time-dependent cytotoxic effect, accompanied by the activation of Akt, ERK1/2, and AMPK, elevated levels of the apoptosis regulator Bcl-2, and a reduction in cellular ATP and mitochondrial membrane potential. The effects of gemigliptin were synergistic with those of the histone deacetylase inhibitor PXD101, the heat-shock protein 90 inhibitor, AUY922, and metformin [138,139,140]. Similarly, sitagliptin has been shown to inhibit the migration, invasion and, at higher concentrations, the growth of thyroid cancer cells in vitro and attenuated tumor growth in a mouse model, possibly by interfering with TGFbeta signaling [60]. In another in vitro study, sitagliptin blocked the growth and clonogenic capacity of gastric cancer cells [141] by inhibiting the YAP transcriptional co-activator. In breast cancer models, DPP-IV has been shown to promote tumorigenesis by enhancing EGF-induced MEK/ERK signaling, leading to AP-1 activation and the expression of the peptidylprolyl cis/trans isomerase encoded by *PIN1*. Sitagliptin suppressed these effects, inhibited colony formation, and induced cytotoxicity by activating apoptotic signaling in vitro [78]. In vitro, sitagliptin also suppressed the growth and migration of endometrial cancer cells [82] and the growth, motility, and invasion of colorectal cancer cells [142].

A general caveat applies to several of the aforementioned studies due to the high gliptin concentrations used and the very limited in vivo data supporting their conclusions. Gliptins effectively inhibit DPP-IV in nanomolar concentrations. The mean tissue concentrations in mice fed sitagliptin are approximately 40 nM [133], the maximum plasma concentrations in humans are below 2 µM with usual dosing (Table 2), and, in a study using repeated doses of up to eight times the usual dose of sitagliptin, maximum plasma concentrations of 11 µM were reported [149]. This contrasts with the millimolar concentrations for which cytotoxicity was reported in most studies (Table 4). While micromolar gliptin concentrations were—by inhibiting DPP-IV—sufficient to enhance the promigratory effect of CXCL12 in chronic myeloid leukemia (CML) cells [147], there was no effect on the growth of DPP-IV-expressing CML cells [147,148]. Similarly, several studies observed no cytotoxicity in DPP-IV-expressing hepatocellular carcinoma cells at concentrations of up to 100 µM [7,67,68,122,143,150], and a recent study using DPP-IV^+^ renal cell carcinoma stem cells demonstrated that sitagliptin had no effect on the growth of these cells, but enhanced their sensitivity to the tyrosine kinase inhibitor, sunitinib, in vitro and in vivo [110]. 

The inhibition of DPP-IV-related proteases may contribute to some of the observed cytotoxic effects. Vildagliptin enhanced the cytotoxicity of a sesquiterpene parthenolide in acute myeloid leukemia cells [145] and was toxic for multiple myeloma cell lines [146]. Similar cytotoxic activity in multiple myeloma was observed for saxagliptin, but not for the more selective DPP-IV inhibitors sitagliptin, alogliptin, and linagliptin. In both studies in hematological malignancies, the anti-tumor effect of vildagliptin was mediated by the inhibition of DPP8 and 9 and not DPP-IV, as demonstrated by knockdown approaches and the use of DPP8 inhibitors [145,146]. Similarly, high concentrations of sitagliptin (above 2 mM) compromised the integrity of the plasma membrane, as evidenced by the LDH release and decreased viability of various cervical carcinoma cell lines, including cells not expressing DPP-IV [144], further supporting the conclusion that these effects are not related to DPP-IV.

The mechanisms responsible for these non-DPP-IV-associated effects of gliptins in cancer cells are currently unknown. Nevertheless, several potential off-target effects were observed for individual gliptins. For example, nanomolar concentrations of linagliptin attenuated cardiac remodeling in DPP-IV-deficient rats, possibly by reducing the TGFbeta-1 and MMP2 expression in cardiac fibroblasts [151]. Another study showed that at clinically relevant concentrations, linagliptin, but not sitagliptin, alogliptin, or saxagliptin, enhanced cardiac recovery after ischemia reperfusion, most likely by stimulating the Akt/eNOS in the endothelial cells, leading to the cyclic guanosine monophosphate (cGMP)-mediated activation of phospholamban, the pivotal regulator of cardiomyocyte contractility [152]. In an ex vivo study, vasodilatory effects mediated by the activation of Kv channels and SERCA pumps were reported for trelagliptin [153]. Further, high concentrations of sitagliptin may affect cells by activating AMPK, as demonstrated in gastric cancer cells [141]. Whether and how these effects of gliptins, seemingly unrelated to DPP-IV inhibition, impact cancer cells remains to be established.

In summary, the evidence for the direct cytotoxic effect of DPP-IV inhibition by gliptins in cancer cells is not convincing.

### 5.2. Effect of DPP-IV Inhibition on Anti-Tumor Immune Response

Anti-tumor immune responses are largely dependent on the infiltration and effector functions of activated T cells. Indeed, subpopulations of CD4^+^ DPP-IV/CD26^high^ T cells seem to have strong anti-tumor activity, possibly due to the fact that DPP-IV/CD26 is a marker of T-cell activation and a co-stimulatory molecule. These cells exhibit a phenotype similar to, but distinct from, Th17 cells, and express several chemokine receptors, cytokines, and anti-apoptotic genes that allow them to migrate, survive, and persist in tumors [154,155]. Since chemokines are key factors driving the infiltration of T cells into the tumor microenvironment, DPP-IV inhibition may modulate anti-tumor immune responses (Figure 1, Table 5). For example, DPP-IV cleaves CXCL10 (IFN-γ-induced protein-10, IP-10), a chemoattractant for various immune cells, and the truncated chemokine acts as an antagonist on its receptor CXCR3 [156]. In a syngeneic melanoma mouse model, tumor growth and the establishment of metastases after the intravenous application of melanoma cells were diminished in DPP-IV knockout animals and after sitagliptin administration. The mechanism of this anti-tumor activity involved higher levels of intact CXCL10 and the increased infiltration of CXCR3-positive lymphocytes into the tumors [157]. As a possible contributing factor, sitagliptin enhanced the recruitment of type 1 conventional dendritic cell precursors (pre-cDC1), which express CXCR3 and are critical for the cross-presentation of exogenous antigens to CD8^+^ T cells [158]. Sitagliptin also delayed tumor growth in a syngeneic colorectal cancer mouse model, which was accompanied by the enhanced infiltration of T and NK cells and improved the response to immunotherapies, including a CpG oligodeoxyribonucleotide, the adoptive transfer of tumor-targeting T cells, and an anti-CTLA-4 immune-checkpoint inhibitor. The combination of sitagliptin with a dual anti-CTLA-4 and anti-PD-1 blockade resulted in tumor rejection in all the experimental animals, whereas a combination therapy with anti-CTLA-4 and anti-PD-1 cured only 42% of the animals [157]. 

A follow-up study [67] by the same group suggests that, at least in certain tumor types, DPP-IV inhibition may also support innate anti-tumor immune responses. In preclinical models of liver and breast carcinoma, the anti-tumor effects of sitagliptin were preserved, even when T cells were depleted. Increased intratumoral concentrations of CCL11, IL4, IL5, and IL33 were observed in sitagliptin-treated animals, and the beneficial effects of sitagliptin were mediated by eosinophils and CCL11, a DPP-IV substrate. Sitagliptin enhanced CCL11-mediated eosinophil infiltration, and the anti-tumor effects seemed to be particularly relevant for tumors producing IL33, a known eosinophil activator and inducer of CCL11 expression. Indeed, anti-IL33 neutralizing antibodies eliminated the beneficial effect of DPP-IV inhibition in liver and breast carcinoma models [67]. The anti-tumor effect of DPP-IV inhibitors in hepatocellular carcinoma was supported by another report, which showed that high intratumoral DPP-IV expression was associated with lower numbers of tumor-infiltrating NK and T cells in patients with hepatocellular carcinoma [68]. In addition, tumor xenotransplant growth was decreased by alogliptin and vildagliptin due to the CXCR3-mediated increase in infiltration by NK cells. Similarly, sitagliptin increased tumor infiltration by NK and T cells in a mouse model of non-alcoholic steatohepatitis-induced hepatocellular carcinoma. Gliptins prevented the degradation of CXCL10 by DPP-IV-expressing cancer cells and preserved its chemotactic activity for NK cells, suggesting that in these models, chemokine-mediated lymphocyte trafficking contributes to the anti-tumor activity of gliptins. Interestingly, the tumors in gliptin-treated animals were also less vascularized [68]. The CXCR3 ligands, CXCL10, CXCL9, and CXCL11, are known to inhibit angiogenesis, but their DPP-IV-mediated truncation seems—in contrast to the effect of truncation on lymphocyte trafficking—not to interfere with their angiostatic properties [176]. The decreased conversion of NPY by DPP-IV to a pro-angiogenic NPY(3-36) Y2/Y5 receptor agonist [177] may be an alternative mechanism, but this remains to be established in future studies.

A study using a syngeneic mouse model of ovarian carcinoma further supports the role of gliptins in enhancing the anti-tumor immune response. Sitagliptin improved animal survival and decreased tumor dissemination in parallel with an increase in intratumoral CD8^+^ T cell/Treg ratio, a decrease in the plasma levels of immunosuppressive cytokines, and an improvement of intratumoral lymphocyte activation and proliferation [159]. The infiltration of the tumors by CXCR3+ T lymphocytes suggested a similar mechanism of immune system activation by sitagliptin, involving the increased bioavailability of chemokines. 

A different mechanism through which vildagliptin leads to the stimulation of the innate anti-tumor immune response was described in lung cancer models. Jang et al. [160] demonstrated that tumor growth was hampered in a syngeneic and xenograft mouse model upon the administration of vildagliptin. However, the T- and B-cell infiltration of the tumors was not substantially different between the treated and untreated animals. By contrast, the macrophages and activated NK cells were much more abundant in the mice receiving vildagliptin, which was accompanied by an increased expression of proinflammatory cytokines, perforin, FasL, and TRAIL. Importantly, the growth-inhibiting effect of vildagliptin was suppressed by macrophage ablation or NK cell depletion, and vildagliptin enhanced TRAIL-mediated lung cancer cell killing by NK cells in vitro. The mechanism behind this immunostimulatory effect proposed by the authors involves the rapid upregulation of surfactant proteins in cancer cells exposed to vildagliptin, which increases the inflammatory activity of macrophages [160].

Although the clinical implications of the aforementioned animal studies remain unclear, a case study reported the rapid regression of hepatocellular carcinoma in a hepatitis C virus-infected patient with poorly controlled diabetes mellitus after sitagliptin was initiated. The residual tumor tissue was heavily infiltrated by CD8^+^ T cells, suggesting the likelihood of an immune-mediated mechanism [178].

Overall, the emerging evidence suggests that, at least in certain tumor types, gliptins may enhance the anti-tumor activity of both the innate and the adaptive immune system, and may act synergistically with currently available immunotherapeutic approaches. Nevertheless, these encouraging preclinical results need to be validated in clinical studies.

### 5.3. Enhancement of Hematopoietic Stem Cell Transplantation and Prophylaxis of Acute Graft-Versus-Host Disease by DPP-IV Inhibition

Hematopoietic stem cell transplantation is currently an important and potentially curative approach in leukemia and lymphoid malignancies and may also be beneficial for some patients with solid tumors [179]. Umbilical cord blood represents an important source of hematopoietic stem/progenitor cells for patients without a HLA-matched sibling or unrelated adult donor. However, delayed engraftment due to a low number of progenitor cells constitutes a significant obstacle. 

Human hematopoietic stem cells express CXCR4, which helps transplanted stem cells to localize to bone marrow niches that actively produce the CXCL12 (SDF-1α) ligand [180,181]. Further, it is now established that the CXCR4–CXCL12 axis is critical in maintaining the bone marrow niche that nurtures HSC, where the disruption of the chemokine signaling pathway results in reduced HSC numbers and reduced ability to repopulate on transplantation [180]. The DPP-IV-processed CXCL12(3-68) has a significantly reduced ability to induce signaling through CXCR4 [26]. Although it has not been observed unequivocally in all studies [182], the genetic or pharmacologic ablation of DPP-IV enzymatic activity both in transplanted cells and in bone marrow enhanced CXCL12-induced responses and improved transplantation efficiency [28,183], implicitly suggesting that the lifetime of CXCL12 was enhanced. In patients with hematological malignancies receiving umbilical cord blood transplants, high-dose sitagliptin was shown to enhance engraftment compared to historic controls at the same institution, and a more sustained inhibition of DPP-IV was associated with improved results [161,162]. An unexpected and intriguing observation from these studies was the very low incidence of graft-versus-host disease, a frequent and significant cause of morbidity and mortality in transplanted patients. This finding stimulated a recently published phase 2 nonrandomized clinical trial on patients who had received myeloablative allogeneic peripheral-blood stem cell transplantation [163]. High-dose sitagliptin (600 mg every 12 h for 15 days) was added to a standard prophylactic regimen consisting of tacrolimus and sirolimus, and the incidence of acute grade II–IV graft-versus-host disease by day 100 was evaluated. None of the 36 patients had toxic side-effects attributable to sitagliptin, all achieved engraftment, and acute graft-versus-host disease developed in two (5.6%) of them, a frequency substantially lower than the 30% incidence reported in similar patient cohorts in the past. The incidences of relapse and chronic graft-versus-host disease were comparable to those in previous studies.

### 5.4. Amelioration of Side-Effects of Conventional Anticancer Treatments by DPP-IV Inhibition

The side-effects of cytotoxic anticancer therapies frequently impact on quality of life and lead to dose reduction or treatment discontinuation in a considerable proportion of cancer patients. Various endogenous DPP-IV substrates, such as glucagon-like peptides, CXCL12, human GM-CSF, and human IL3 [2,3,4,5,6], may have protective effects on the chemotherapy-induced damage to healthy tissues. Combined with the favorable safety profile of gliptins, this was an impetus for several, so far mostly preclinical, studies examining the potential of DPP-IV inhibition to mitigate the side-effects of chemotherapy (Figure 1, Table 6).

In addition to the importance of DPP-IV in modulating the CXCL12–CXCR4 axis, studies on the role of DPP-IV in hematopoietic stem cell engraftment revealed that DPP-IV may cleave GM-CSF, G-CSF, IL3, and erythropoietin and decrease their activity [164]. Furthermore, mice receiving sitagliptin prior to and shortly after 5-fluorouracil administration had higher cellularity in the bone marrow and greatly accelerated hematopoietic progenitor cell recovery, as evidenced by the increased colony formation and number of immunophenotypically defined long-term and short-term hematopoietic stem cells. The enhancement of the recovery in the peripheral blood by sitagliptin was modest and was not seen in mice that received sitagliptin only after the administration of 5-fluorouracil, hinting that the timing, dosing, and duration of gliptin administration plays an important role. Similar, albeit more pronounced effects were seen in DPP-IV knockout mice, supporting the conclusion that the absence of DPP-IV enzymatic activity facilitates the recovery of hematopoiesis after chemotherapy [164]. 

Various gliptins seem to have nephroprotective effects. In a rat model, the administration of teneligliptin attenuated cisplatin-induced acute kidney injury and accelerated the recovery of renal functions [165]. A histopathological analysis of the kidney tissues of animals treated with a DPP-IV inhibitor revealed the decreased apoptosis of the proximal tubule epithelial cells in the early stages and, subsequently, increased proliferation at a later stage, decreased tubulointerstitial fibrosis, and increased infiltration with reparatory M2 macrophages. In vitro, tenaligliptin increased the proliferation of proximal tubule epithelial cells mediated by CXCL12–CXCR4 signaling, suggesting a possible mechanism for its beneficial effects on cisplatin-induced nephrotoxicity [165]. Another DPP-IV inhibitor, alogliptin, has been shown to have protective effects in a rat model of cyclophosphamide-induced nephrotoxicity [166]. The coadministration of alogliptin improved the functional and histopathological markers of renal injury. An analysis of the possible mechanisms underlying these protective effects revealed that alogliptin diminished the renal levels of TNFalpha and TGFbeta and the activation of JNK1 and SMAD signaling, and improved the markers of oxidative stress. In addition, the activation of caspase-3 and the Bax/Bcl-2 ratio were significantly reduced by alogliptin. It remains to be confirmed whether the increased bioavailability and signaling of GLP-1 proposed by the authors [166] represent a unifying molecular mechanism of these effects of DPP-IV inhibition. The anti-inflammatory effects of gliptins were proposed to contribute to their ability to diminish doxorubicin nephrotoxicity in rats. Sitagliptin and linagliptin decreased tubulointerstitial injury and interstitial fibrosis, possibly by reducing the activation of the NOD-like receptor containing pyrin domain 3 (NLRP3) inflammasome [167]. In line with these findings, an independent study reported similar beneficial effects of vildagliptin and saxagliptin on kidney function, the kidney expression of TNF-alpha, IL1beta, neutrophil gelatinase-associated lipocalin (NGAL), and NLRP3 and morphological alterations induced by doxorubicin [168].

The protective effect of GLP-1 and GLP-2 in chemotherapy-induced alimentary mucositis [184] led to the hypothesis that gliptins may prevent this common complication of chemotherapy. Indeed, the administration of vildagliptin attenuated the severity of 5-fluorouracil-induced diarrhea, morphological changes in the small intestine, and TNF-alpha expression [175] in a mouse model. GLP-1 and gliptins were also proposed as neuroprotective agents [185,186]. Alogliptin ameliorated oxaliplatin-induced mechanical allodynia and axonal degeneration in rats and prevented neurite shortening in vitro [172]. Interestingly, these effects were selective for platinum-based chemotherapeutics, as there were no effects on paclitaxel- or bortezomib-induced neuropathy. The molecular mechanism remains to be established, but the in vitro results of this study suggested that GLP-1 receptor signaling was not involved [172]. Another study on rats demonstrated that a 10-day treatment with sitagliptin prior to the administration of doxorubicin ameliorated ECG changes, lowered the level of serum markers of cardiotoxicity, and improved histopathological alterations in the heart. This was accompanied by decreased lipid peroxidation, increased levels of superoxide dismutase and reduced glutathione, and decreased markers of inflammation and apoptosis in cardiac tissue [170]. Similarly, linagliptin treatment ameliorated the cardiotoxicity induced by the repeated administration of lower doses of doxorubicin and reduced lipid peroxidation [171]. An improved antioxidant status after gliptin administration was also observed in methotrexate-induced hepatic injury [173]. The pretreatment of mice with sitagliptin for five days before methotrexate administration reduced the elevation of the serum transaminases, ALP and LDH, improved histopathological changes in the liver, and was accompanied by the alleviation of lipid peroxidation and the preservation of the antioxidants superoxide dismutase and glutathione. The expression of Nrf2, a transcription factor regulating the expression of antioxidant enzymes [187], was preserved in animals pretreated with sitagliptin, whereas methotrexate suppressed it. Sitagliptin also counteracted the activation of NF-kappaB signaling and the expression of proinflammatory mediators in the liver and reduced the hepatocyte apoptosis induction caused by methotrexate administration [173]. A recent study further demonstrated that linagliptin substantially alleviates testicular injury after the administration of cisplatin. Increased intratesticular levels of CXCL12 and the abrogation of the beneficial effects of linagliptin by a CXCR4 antagonist suggested enhanced CXCL12–CXCR4 signaling as the underlying mechanism [174]. 

A possible limitation of individual preclinical studies may be that most of them utilize only a single DPP-IV inhibitor. Nevertheless, the overall findings of the published studies using various gliptins supports the conclusion that the underlying mechanism involves the inhibition of DPP-IV enzymatic activity. Nevertheless, the identification of the critical endogenous DPP-IV substrate(s) and the molecular mechanisms in individual organs remains an important objective. The lack of clinical data is another limitation. Recently, a small retrospective study on cancer patients with diabetes treated with high-dose cisplatin supported the preclinical findings, revealing that patients treated with gliptins had lower incidences of acute kidney injury and a smaller decline in estimated glomerular filtration rate after treatment with cisplatin compared to other antidiabetic drugs [169]. A randomized double-blind placebo-controlled trial to evaluate the effect of gliptins on cisplatin toxicity in cancer patients was initiated [188], but to date, no results have been published in the literature.

## 6. Conclusions and Unanswered Questions

DPP-IV inhibition is currently a standard therapeutic approach in patients with type 2 diabetes mellitus. Gliptins, specific, orally administered DPP-IV inhibitors with a favorable side-effect profile, do not only lower glycemia, but appear to have several beneficial effects in diabetic patients [14,16]. Despite the lack of consensus in the field, the studies summarized in this review suggest that in selected situations, gliptins may also be of benefit for patients at risk of or suffering from cancer. DPP-IV inhibition may have chemopreventive effects, especially in the context of hepatocellular carcinoma. From this point of view, the use of gliptins may seem advantageous, particularly for diabetic patients with NASH, an increasingly important predisposing factor for hepatocellular cancer in the Western world. However, long-term studies are needed to verify the clinical utility of gliptins in this respect. The neutral effect of gliptins on the incidence of colorectal carcinoma in epidemiological studies [125] and the potential beneficial effects observed in various preclinical models [129,130,131] are also encouraging. Colorectal cancer ranks among the most common cancers in both sexes, and its incidence is increased in patients with diabetes. Various studies, which are mostly preclinical at this point, further demonstrate that gliptins may be used in new applications in the context of cancer treatment (Figure 1).

The evidence is rather limited for the notion that gliptins may elicit cytotoxicity in cancer cells by inhibiting DPP-IV enzymatic activity. With some exceptions [60,64], in vivo studies that support direct cytotoxicity are currently lacking, and the majority of the in vitro studies reporting cytotoxic effects used concentrations several orders of magnitude higher than those achievable in clinical practice. By contrast, increased engraftment and the decreased incidence of acute graft-versus-host disease [161,162,163] in patients undergoing hematopoietic stem cell transplantation receiving sitagliptin are supported by preclinical mechanistic and advanced clinical-stage studies. These promising results warrant validation in placebo-controlled trials in larger patient cohorts. Based on the preclinical data showing improved response to immunotherapies [67,68,157,159,160], gliptins were included in ongoing clinical trials utilizing checkpoint inhibitors (e.g., linagliptin in NCT03337698 and NCT03281369). The preclinical data also suggest that DPP-IV inhibitors may aid in mitigating the side-effects of currently used anticancer drugs—nephroprotective [165,166,167,168], neuroprotective [172], and hepatoprotective [173] effects, and beneficial effects on cardiotoxicity [170] and mucosal injury [175] have been reported. 

Nevertheless, several factors may complicate the clinical application of gliptins in cancer patients and need to be taken into account when designing larger studies and randomized trials. The precise molecular mechanisms linking DPP-IV inhibition and various beneficial effects, including the prevention of cancer initiation and the protection of healthy tissues from the toxic effects of chemotherapy, remain largely unknown, but likely include the increased signaling of DPP-IV substrates, such as CXCL12. The same substrates may nevertheless contribute to the development of protective niches for cancer cells and thus lead to therapy resistance. In vitro studies cannot fully address this concern, and there is only limited evidence from animal models that the anti-tumor effect of cytostatic drugs is not affected by the concomitant application of gliptins [172]. It has been established that leukemic stem cells (LSC) in CML express CXCR4 and utilize the CXCL12–CXCR4 signaling pathway; CXCL12 from mesenchymal stromal cells helps to maintain the LSC in a quiescent G_0_ state, which renders these BCR-ABL^+^ LSC resistant to tyrosine kinase inhibitors (TKI). The inhibition of stromal cell CXCL12 releases the cells from the niche constraints and the LSCs enter the cell cycle, self-renew, and become TKI-sensitive [189]. Clearly, the inhibition of DPP-IV activity in this circumstance might be counter-productive, enhancing the ability of CXCL12 to maintain the mesenchymal stromal cell niche. With this precedent in mind, it is important to note that the niche CXCL12–CXCR4 axis is also utilized by acute myeloid leukemia cells in the bone marrow [190] and may be more generally applicable to most leukemias [191]. In addition, bone-derived CXCL12 may have systemic effects, promoting the growth and dissemination of primary breast tumors and facilitating their metastasis to the bones [192]. Similar growth- and metastasis-promoting effects of the CXCL12–CXCR4 axis were also observed in other cancer types [73,193]. Thus, the use of gliptins seems problematic in tumors that hijack the CXCL12–CXCR4 axis, especially those that simultaneously express DPP-IV. By protecting CXCL12 and other bioactive substrates from DPP-IV cleavage, gliptins may also promote the release, homing, and angiogenic capacity of bone marrow-derived progenitor cells [194]. While this increased angiogenic potential may be beneficial for patients with diabetes or myocardial infarction, for example [195], it may be detrimental for patients with cancer. Supporting the potential effect of DPP-IV inhibition on newly formed vasculature, a study demonstrated that gliptins may increase blood vessel permeability via the CXCL12–CXCR4 axis [196]. The consequences of these findings in oncology are currently unclear, but dysfunctional vasculature is a known contributor to both tumor progression and resistance to therapy [197].

When considering the potential use of DPP-IV inhibition in modulating the immune response to cancers, several aspects rise to the fore. First, CD26 is differentially expressed on immune cells—would the inhibition of the DPP-IV activity enhance the potential anti-tumor activity of each subset, or would the effect depend on the cell type? [198,199]. Second, do the tumor cells themselves, and/or other stromal cells in the tumor, express DPP-IV, and, if so, would the inhibition of the expressed DPP-IV activity potentiate or inhibit local immune function, perhaps by modulating the chemokine-driven attraction of immune cells? [53,200]. In addressing these issues, an important consideration is that animal models may offer limited insight, as CD26/DPP-IV is more broadly expressed in immune cells in mice compared to humans, where it is predominantly expressed in T cells [33,35,50]. Mucosal-associated invariant T (MAIT) cells express high levels of CD26 and are critical regulators of commensal organisms at the mucosal interface, with regulatory roles in tissue repair, inflammation, and autoimmune regulation [201]. Given the restricted recognition potential and tissue location, even though DPP-IV inhibition may affect MAIT function, it would be difficult to establish a significant role for MAIT in potentiating or inhibiting tumor development. On the other hand, two studies described the anti-tumor activity of CD26 high-CD4^+^ T cells, but it is currently unclear whether, how, or at which stage DPP-IV/CD26 may play a role in the development of this T-cell subpopulation [154,155]. The loss of CD26 did not affect the cytokine production or the anti-tumor effect of these cells [155], suggesting that the functionality of this subpopulation is not affected by DPP-IV inhibitors.

Two mechanisms may oppose the possible regulatory function of DPP-IV for chemokines directing tissue infiltration by immune cells. First, as a scenario that may offer part of the explanation for the differing efficacies or the lack of effect of DPP-IV inhibitors as a therapeutic modality, tumor microenvironments differ in terms of the extracellular matrix molecular composition, rigidity, and activity of local metalloproteases, rendering the area inaccessible to the potential infiltration of NKT or activated T cells [202,203]. This set of circumstances would negate any effect of DPP-IV inhibitors on chemokine longevity. Secondly, following processing by DPP-IV, the truncated chemokines CCL5(3-68)/RANTES, CCL3L1(3-70), and CCL4(3-69)/MIP1beta increase their ligand-binding affinity and signaling potential for their respective cognate receptors expressed on T cells and polymorphonuclear leukocytes. DPP-IV inhibition in this chemokine environment might reduce T-cell recruitment to potentially inflammatory sites [11,45]. More generally, however, the inhibition of the DPP-IV processing of susceptible chemokines is likely to potentiate the activity of these chemokines and lead to increased infiltration into inflammatory sites. If these sites are intra-tumoral, this seems likely to be beneficial.

It remains to be established whether compounds that, in addition to DPP-IV, inhibit other DPP-IV-like proteases, such as FAP, DPP8, and DPP9, may offer advantages over the use of more selective DPP-IV inhibitors in patients with cancer. Talabostat, a pan-dipeptidyl peptidase inhibitor, was initially tested with encouraging preclinical results in a broad spectrum of tumors, but this did not translate into beneficial effects in phase II trials in combination with chemotherapy (for review see [204]). Nevertheless, a recent study shows that its combination with immunotherapy may be more suitable. Talabostat inhibited tumor growth in pancreatic ductal adenocarcinoma models and promoted the infiltration of the tumors by CXCR3^+^ T and NK cells, most likely by inhibiting DPP-IV. In addition, probably by inhibiting DPP8/9, it activated the inflammasome and pyroptosis. These proinflammatory effects synergized with anti-PD1 treatments in a T- and NK-cell-dependent manner and led to an effective anti-tumor response that protected the cured animals against re-challenge with tumor cells [205]. Similar pro-inflammatory effects, leading to the increased infiltration of CD8^+^ cells and activation of the inflammasome, were also observed with another pan-DPP inhibitor, ARI-4175, in hepatocellular carcinoma models [69]. 

In summary, our increased understanding of the role of DPP-IV enzymatic activity in regulating incretins and other biopeptides may open the possibility of repurposing gliptins for use in the field of oncology. The modulation of DPP-IV activity is almost certainly not going to be a critical component of anti-tumor treatment in general, but may be of benefit in selected tumor types and circumstances. The identification of these specific circumstances remains an important task for ongoing preclinical and clinical studies.

## Figures and Tables

**Figure 1 cancers-14-02072-f001:**
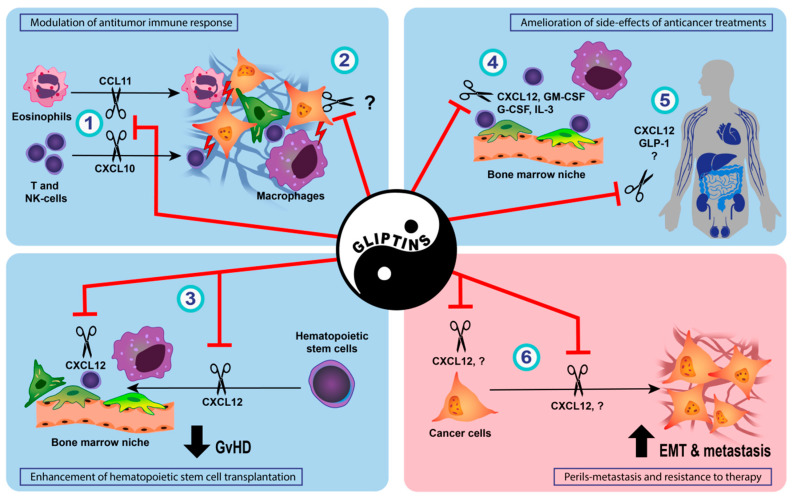
Potential utility and pitfalls of DPP-IV inhibition by gliptins in anticancer treatment. (1) By preventing DPP-IV-mediated cleavage of chemokines, gliptins may support the recruitment of immune cells with anti-tumor activity into the tumor microenvironment [67,68,157,159]. (2) In addition, changes in cancer cells induced by gliptins may activate macrophages and NK cells [160]. (3) In patients undergoing hematopoietic stem cell transplantation, attenuated cleavage of CXCL12 facilitates the homing and engraftment of donor cells [28,161,162] and result in lower incidence of acute graft-versus-host disease (GvHD) [163]. By reducing the DPP-IV-mediated cleavage of various mediators supporting tissue regeneration, gliptins may increase the resilience of healthy tissues to chemotherapy-induced damage, resulting in (4) accelerated hematopoietic recovery after chemotherapy [164], (5) decreased nephrotoxicity [165,166,167,168,169], cardiotoxicity [170,171], neurotoxicity [172], hepatotoxicity [173], testicular toxicity [174], and reduced mucositis [175]. (6) On the other hand, increased bioavailability of CXCL12 resulting from DPP-IV inhibition, together with gliptin-induced activation of nuclear factor E2–related factor 2 (Nrf2), carries the risk of accelerated epithelial–mesenchymal transition (EMT) and metastatic dissemination of cancer cells [7,8,58,73,80].

**Table 1 cancers-14-02072-t001:** Examples of bioactive peptides and proteins cleaved by DPP-IV [2,3,4,5,6].

Chemokines	CXCL12 (SDF-1), CXCL11 (I-TAC), CXCL10 (IP-10), CXCL9 (Mig), CCL11 (eotaxin), CCL5 (RANTES), CCL3L1, CCL22 (MDC)
Incretins	GLP-1, GIP
Neuropeptides	SP, PYY, NPY, VIP, PACAP
Other	GLP-2, BNP, erythropoietin, GHRH, GRP, glucagon, procalcitonin

BNP = Brain natriuretic peptide; GHRH = Growth-hormone-releasing hormone; GIP = Glucose-dependent insulinotropic polypeptide; GLP = Glucagon-like peptide; GRP = Gastrin-releasing peptide; IP-10 = Interferon-gamma-induced protein 10; I-TAC = Interferon-inducible T-cell alpha chemoattractant; MDC = Macrophage-derived chemokine; Mig = Monokine induced by gamma interferon; NPY = Neuropeptide Y; PACAP = Pituitary adenylate cyclase-activating polypeptide; PYY = Peptide YY; RANTES = Regulated on activation, normal T cell expressed and secreted; SDF-1 = Stromal-cell-derived factor 1; SP = Substance P; VIP = Vasoactive intestinal peptide.

**Table 2 cancers-14-02072-t002:** Selected pharmacodynamic and pharmacokinetic properties of gliptins evaluated in cancer studies [21,22,23,24,25].

Generic Name	Daily Dose (mg) Used in Diabetic Patients	IC50 DPP-IV (nM)	IC50 DPP8 (nM)	IC50 DPP9 (nM)	IC50 FAP (nM)	Cmax (nM) in Humans
Gemigliptin	50	6.3	277,000	233,000	418,430	128
Sitagliptin	100	19	48,000	>100,000	>500,000	959
Vildagliptin	100 (2 × 50)	3	810	95	54,600	1309
Saxagliptin	5	1.3	520	98	>1000	76
Linagliptin	5	1	>100,000	>100,000	89	10
Alogliptin	25	6.9	>100,000	>100,000	>500,000	324
Teneligliptin	20	0.37	260	540	>10,000	645
Anagliptin	200 (2 × 100)	3.8	63,000	60,000	72,700	1242

**Table 4 cancers-14-02072-t004:** Summary of studies evaluating direct cytotoxic effects of gliptins.

Tumor Type	Gliptin	Effective Gliptin Concentrations in In Vitro Studies	References, Notes
Colorectal cancer	Vildagliptin	Cytotoxicity: 2–10 mM for single exposure and >0.328 mM for repeated exposure. Reduced expression of EMT markers: 10–20 µM Reduced cyclin-dependent kinase 1 phosphorylation: 0.08–0.16 mM	Suppression of lung metastases also observed in an animal model [64].
	Sitagliptin	Cytotoxicity: above 0.5 mM, lower concentrations did not substantially influence cell growth. Inhibition of motility and invasion: 0.5 mM	[142]
Thyroid cancer	Gemigliptin	Cytotoxicity: 0.5–2 mM Effect in other assays (reduced migration, induction of apoptosis etc.): 0.25–1 mM. At this concentration, some cytotoxicity was seen in normal bronchial epithelial cells.	Synergistic cytotoxic effects with a histone deacetylase inhibitor, metformin, and a Hsp90 inhibitor [138,139,140].
	Sitagliptin, vildagliptin	Reduced cell growth: 1 mM Decreased migration and invasion: 10–100 nM	Decreased tumor growth was also observed in a xenotransplantation mouse model [60].
Gastric cancer	Sitagliptin	Inhibition of growth and colony formation: 1–2 mM Inhibition of YAP signaling: 1–2 mM	[141]
Breast cancer	Sitagliptin	Decreased cell viability and activation of apoptotic signaling: 0.5–2.5 mM Decreased colony formation: 0.1–1 mM Inhibition of EGF signaling: 0.5–1 mM	Cells pretreated with sitagliptin (10 mM) form smaller tumors in experimental animals [78].
Endometrial cancer	Sitagliptin	Decreased cell growth: 2–8 mM Decreased migration: 1 mM	[82]
Hepatocellular carcinoma	Sitagliptin	No effect on cell growth or synergy with doxurubicine derivative WP 631 toxicity: 0.01–200 µM	[143]
Cervical carcinoma	Sitagliptin	Compromised cellular integrity (LDH release): >2 mM Decreased viability: >0.2 mM Decreased cell adhesion 1 mM	Effects are independent of DPP-IV expression [144].
Acute myeloid leukemia	Vildagliptin	10 µM vildagliptin but not 10 µM sitagliptin enhances the cytotoxic effect of parthenolide.	Effects caused by DPP8 and DPP9 inhibition [145].
Multiple myeloma	Vildagliptin, saxagliptin	Cytotoxicity: 0.005–0.1 mM	Effects caused by DPP9 inhibition [146].
Chronic myeloid leukemia	Vildagliptin, sitagliptin, saxagliptin	No effect on cell growth or synergy with tyrosinkinase inhibitors: 10 nM–10 µM Reduced mobilization of leukemic stem cells from a stroma cell layer: 5–10 µM	Gliptins enhance SDF-1 induced migration, but do not affect colony formation. Preincubation with vildagliptin decreased engraftment of leukemic cells in mice. Gliptin treatment led to decreased BCR/ABL1 transcript levels in two patients [147]. Effect on engraftment was not confirmed in a follow-up study [148].

**Table 5 cancers-14-02072-t005:** Summary of studies suggesting antitumor effect of gliptins mediated by promotion of antitumor immune response.

Tumor TYPE	Gliptin	Proposed Mechanism	Notes, Reference
Melanoma, colorectal carcinoma	Sitagliptin	Preserved bioactivity of CXCL10, leading to increase CXCR3-dependent infiltration of CD4^+^ and CD8^+^ lymphocytes.	No effect of sitagliptin on tumor growth in DPP-IV KO mice suggests that sitagliptin does not have a direct cytotoxic effect [157].
Hepatic cancer	Anagliptin, vildagliptin, sitagliptin	Preserved bioactivity of CXCL10, leading to increase CXCR3-dependent infiltration of NK cells.	Gliptins do not affect the growth of hepatocellular carcinoma cells in vitro (up to 100 µM) [68].
Hepatic and breast cancer	Sitagliptin	Preserved bioactivity of CCL11, leading to increased infiltration of eosinophils.	No effect of sitagliptin on hepatic carcinoma cell growth (up to 12.3 µM) [67].
Ovarian cancer	Sitagliptin	Infiltration of the tumors by CXCR3^+^ T lymphocytes	[82]
Lung cancer	Vildagliptin	Increased expression of surfactant proteins in cancer cells, resulting in higher amounts and pro-inflammatory activity of macrophages and NK cells.	The antitumor effect of vildagliptin was preserved in CD26^−/−^ animals. No cytotoxicity observed for 0.3–1.3 mM vildagliptin in vitro, increased surfactant protein expression after treatment with 10–20 µM vildagliptin [160].

**Table 6 cancers-14-02072-t006:** Summary of studies suggesting amelioration of side-effects of chemotherapy by gliptins.

Observed Effect	Gliptin	Anticancer Drug	Proposed Mechanism	Notes, Reference
Reduced myelotoxicity	Sitagliptin	5-fluorouracil	Decreased cleavage of GM-CSF, G-CSF, and IL3, leading to increases in recovery of hematopoietic progenitor cells and bone marrow cellularity.	Similar effects observed in CD26^−/−^ mice [164]
Nephroprotection	Teneligliptin	Cisplatin	Possible anti-inflammatory effects and inhibition of CXCL12 breakdown.	[165]
	Alogliptin	Cyclophosphamide	Reduced oxidative stress and inflammation.	[166]
	Sitagliptin, Linagliptin	Doxorubicin	Decreased expression of NLRP3 inflammasome-associated genes.	[167]
	Vildagliptin, Saxagliptin	Doxorubicin	Decreased inflammation.	[168]
Decreased mucositis	Vildagliptin	5-fluorouracil	Possibly preserved bioactivity of GLP-1 and 2.	[175]
Neuroprotection	Alogliptin	Oxaliplatin	Unknown.	Effect seen in oxaliplatin-induced, but not bortezomib- or paclitaxel-induced neuropathy [172]
Reduced cardiotoxicity	Sitagliptin	Doxorubicin	Reduced oxidative damage, inflammation, and apoptosis in cardiac tissue.	[170]
	Linagliptin	Doxorubicin	Decreased oxidative stress.	[171]
Hepatoprotection	Sitagliptin	Methotrexate	Reduced oxidative stress and inflammation.	[173]
Reduced reproductive toxicity	Linagliptin	Cisplatin	Increased bioactivity of CXCL12.	[174]

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
