# Peer review of "Does DPP-IV Inhibition Offer New Avenues for Therapeutic Intervention in Malignant Disease?"

_cancers, 2022, doi:10.3390/cancers14092072_

Round 1

Reviewer 1 Report

This is an outstanding, thorough and much needed balanced review of the field of DPP4 inhibition in cancer. Well written by veterans of the DPP4 field. Congratulations to the authors.

Overall, it indicates that the weight of sound literature points to a plethora of likely and potential benefits for prevention and co-therapy of cancers, with very low or no cytotoxicity from DPP4 inhibition .

Page 4.   It should be clear in this document that the mouse-human differences mentioned include that fibronectin has been shown to bind mouse but not human DPP4, and that human but not mouse DPP4 binds to ADA. Indeed, the binding sites used by mouse DPP4 for fibronectin and human DPP4 for ADA appear to be in almost the same location on the DPP4 molecule.

The literature cited is impressively massive. However, a few additional papers should be considered:

  1. This paper is relevant to the discussion of CXCR3 mediated chemotaxis in melanoma:
    Stuart J. Cook, etal 2018 Differential chemokine receptor expression and usage by pre-cDC1 and pre-cDC2. Immunol Cell Biol96(10): 1131-9. doi 10.1111/imcb.12186 PMID 29920767
  2. Page 7 line 323: This paper is relevant:

 XM Wang, et al 2017 Profibrotic role of dipeptidyl peptidase 4 in carbon tetrachloride induced experimental liver injury. Immunol Cell Biol95(5): 443-53. doi 10.1038/icb.2016.116 PMID 27899813

  1. Section 5.2: Please consider the body of work described here for addressing the question of a role for DPP4 in angiogenesis:
  2. E. Kuo, K. Abe, Z. Zukowska 2007 Stress, NPY and vascular remodeling: Implications for stress-related diseases. Peptides, 28(2): 435-40. doi    10.1016/j.peptides.2006.08.035 PMID 1868452

Page 4 line 149: Ref 43 is also applicable here.

Author Response

Thank you for the positive and kind evaluation and valuable suggestions. Point by point response (reviewers’ comments in italics):

This is an outstanding, thorough and much needed balanced review of the field of DPP4 inhibition in cancer. Well written by veterans of the DPP4 field. Congratulations to the authors.

Overall, it indicates that the weight of sound literature points to a plethora of likely and potential benefits for prevention and co-therapy of cancers, with very low or no cytotoxicity from DPP4 inhibition.

Page 4.   It should be clear in this document that the mouse-human differences mentioned include that fibronectin has been shown to bind mouse but not human DPP4, and that human but not mouse DPP4 binds to ADA. Indeed, the binding sites used by mouse DPP4 for fibronectin and human DPP4 for ADA appear to be in almost the same location on the DPP4 molecule.

On page 4, it is now specifically stressed that only human, but not mouse DPP-IV can bind ADA and that data showing binding of human DPP-IV to fibronectin have to the best of our knowledge not been published. We are aware of the literature demonstrating fibronectin binding in experiments using rodent DPP-IV/cells. One study suggested that binding of (rat) DPP-IV to (human) fibronectin is lower compared to its binding to collagens (Loster, K., et al., The cysteine-rich region of dipeptidyl peptidase IV (CD 26) is the collagen-binding site. Biochem Biophys Res Commun, 1995. 217(1): p. 341-8.) and that soluble form of human DPP-IV may have limited capacity to bind collagen (Gorrell, M.D., V. Gysbers, and G.W. McCaughan, CD26: a multifunctional integral membrane and secreted protein of activated lymphocytes. Scand J Immunol, 2001. 54(3): p. 249-64. ). One study speculated a possible role of DPP-IV-fibronectin interaction in human blastocyst implantation, but the evidence was correlative (Shimomura, Y., et al., Possible involvement of crosstalk cell-adhesion mechanism by endometrial CD26/dipeptidyl peptidase IV and embryonal fibronectin in human blastocyst implantation. Mol Hum Reprod, 2006. 12(8): p. 491-5).

The literature cited is impressively massive. However, a few additional papers should be considered:

1.This paper is relevant to the discussion of CXCR3 mediated chemotaxis in melanoma:
Stuart J. Cook, etal 2018 Differential chemokine receptor expression and usage by pre-cDC1 and pre-cDC2. Immunol Cell Biol,  96(10): 1131-9. doi 10.1111/imcb.12186 PMID 29920767

Reference to Cook et al 2018 was included as a potential mechanism leading to more effective antitumor immune response.                                                     

2. Page 7 line 323: This paper is relevant:

 XM Wang, et al 2017 Profibrotic role of dipeptidyl peptidase 4 in carbon tetrachloride induced experimental liver injury. Immunol Cell Biol,  95(5): 443-53. doi 10.1038/icb.2016.116 PMID 27899813

Reference to Wang et al 2017 was included.

3. Section 5.2: Please consider the body of work described here for addressing the question of a role for DPP4 in angiogenesis:

4. E. Kuo, K. Abe, Z. Zukowska 2007 Stress, NPY and vascular remodeling: Implications for stress-related diseases. Peptides, 28(2): 435-40. doi    10.1016/j.peptides.2006.08.035 PMID 1868452]

Reference to the possible antiangiogenic effect of gliptins due to decreased conversion of NPY to NPY(3-36) was added. The possible role of the DPP-IV-mediated cleavage of NPY in hepatocellular cancer was also included in part 3 (Dietrich, P.; Wormser, L.; Fritz, V.; Seitz, T.; De Maria, M.; Schambony, A.; Kremer, A.E.; Gunther, C.; Itzel, T.; Thasler, W.E.; et al. Molecular crosstalk between Y5 receptor and neuropeptide Y drives liver cancer. J. Clin. Invest. 2020, 130, 2509-2526, doi:10.1172/JCI131919.). A further reference to possible angiogenesis modulating effects of DPP-IV inhibition was added to part 6.

Page 4 line 149: Ref 43 is also applicable here.

Reference to Yu et al 2011 was included.

Sincerely yours,

Petr Busek (on behalf of all authors)

Reviewer 2 Report

Numerous studies have demonstrated conflicting functions of DPP-IV across various cancers or tissues. A similar controversy exists for DPP-IV inhibitors. This review performed a comprehensive summary of published studies, provided a deep discussion of opposing observations, and highlighted the potential utility and pitfalls of DDP-IV inhibitors in treating tumors.

  1. Should add at least one sentence in the main text to describe table 1
  2. This manuscript focuses on cancers which should also be reflected in the title. Malignant disease is a more general term.
  3. The title of section 5.1 is “Direct anti-tumor effects of DPP-IV inhibition”, however, the second half of section 4 is also about anti-tumor effects. So the authors may consider changing the title of section 5.1, for example, to include “cytotoxic effect”
  4. I have no idea why there is a “Scheme 12” under 5.4. This paragraph should be placed in a more appropriate place.
  5. Figure 1 is a clear representation of what’s been discussed in the text. But there is no figure legend or description for it.

Author Response

Thank you for the positive evaluation and valuable suggestions. Point by point response (reviewers’ comments in italics):

Numerous studies have demonstrated conflicting functions of DPP-IV across various cancers or tissues. A similar controversy exists for DPP-IV inhibitors. This review performed a comprehensive summary of published studies, provided a deep discussion of opposing observations, and highlighted the potential utility and pitfalls of DDP-IV inhibitors in treating tumors.

  1.  Should add at least one sentence in the main text to describe table 1

Reference to Table 1 is on page two, we added references to review articles detailing the pleiotropic biological functions of the proteins and the role of DPP-IV.

„Its proteolytic activity cleaves an X-Pro or X-Ala dipeptide from the N terminus of various peptides and proteins comprising a large group of bioactive molecules with pleiotropic biological functions (Table 1, see [2-6] for review).“

2. This manuscript focuses on cancers which should also be reflected in the title. Malignant disease is a more general term.

The final version of the title was created in collaboration with our native speaking co-author (JDC). We agree that several other (non-malignant) illnesses and states can be very detrimental and considered in this respect “malignant”. However, the term “malignant disease“ seems to be synonymous with cancer, see e.g. https://www.britannica.com/science/history-of-medicine/Malignant-disease, https://www.ncbi.nlm.nih.gov/books/NBK82159/ https://clinicalgate.com/malignant-disease-2/. Based on this, we would prefer to preserve the original title and we believe it appropriately represents the focus of the review.

3. The title of section 5.1 is “Direct anti-tumor effects of DPP-IV inhibition”, however, the second half of section 4 is also about anti-tumor effects. So the authors may consider changing the title of section 5.1, for example, to include “cytotoxic effect”

Thank you for the suggestion, the heading of this section was amended.

4. I have no idea why there is a “Scheme 12” under 5.4. This paragraph should be placed in a more appropriate place.

Thank you for spotting this mistake- part of the text disappeared during final formatting of the text. Amended.

5. Figure 1 is a clear representation of what’s been discussed in the text. But there is no figure legend or description for it.

Amended- figure legend should be now clearly separated from the main text.

Once again thank you for your suggestions.

Sincerely yours,

Petr Busek (on behalf of all authors)